# Error Function Optimization to Compare Neural Activity and Train Blended Rhythmic Networks

**DOI:** 10.3390/brainsci14050468

**Published:** 2024-05-07

**Authors:** Jassem Bourahmah, Akira Sakurai, Andrey L. Shilnikov

**Affiliations:** 1Neuroscience Institute, Georgia State University, 100 Piedmont Ave., Atlanta, GA 30303, USA; jbourahmah1@student.gsu.edu; 2Department of Mathematics & Statistics, Neuroscience Institute, Georgia State University, 100 Piedmont Ave., Atlanta, GA 30303, USA; akira@gsu.edu

**Keywords:** CPG, blended approach, neural network, parameter optimization, model training

## Abstract

We present a novel set of quantitative measures for “likeness” (error function) designed to alleviate the time-consuming and subjective nature of manually comparing biological recordings from electrophysiological experiments with the outcomes of their mathematical models. Our innovative “blended” system approach offers an objective, high-throughput, and computationally efficient method for comparing biological and mathematical models. This approach involves using voltage recordings of biological neurons to drive and train mathematical models, facilitating the derivation of the error function for further parameter optimization. Our calibration process incorporates measurements such as action potential (AP) frequency, voltage moving average, voltage envelopes, and the probability of post-synaptic channels. To assess the effectiveness of our method, we utilized the sea slug *Melibe leonina* swim central pattern generator (CPG) as our model circuit and conducted electrophysiological experiments with TTX to isolate CPG interneurons. During the comparison of biological recordings and mathematically simulated neurons, we performed a grid search of inhibitory and excitatory synapse conductance. Our findings indicate that a weighted sum of simple functions is essential for comprehensively capturing a neuron’s rhythmic activity. Overall, our study suggests that our blended system approach holds promise for enabling objective and high-throughput comparisons between biological and mathematical models, offering significant potential for advancing research in neural circuitry and related fields.

## 1. Introduction

### Central Pattern Generators

A central pattern generator (CPG) comprises interneurons that generate stable rhythmic patterns from non-rhythmic input. Such multi-phase patterns or repetitive rhythms are critical for animal functioning and behavior [1,2]. Examples of motor actions controlled by CPGs include heartbeat, respiration, sleep, chewing, and locomotion [3,4,5,6,7,8,9,10]. CPGs have also been suggested as the operational machinery of the cortex [11]. There are two basic types of CPG models and their building blocks: biophysically realistic and phenomenological. Biophysical models incorporate known molecular mechanisms underlying the generation of electrical activity in neurons and synapses, while phenomenological models simplify the firing of an action potential (AP) as an instantaneous pulse and effective synaptic coupling [12,13,14,15,16,17].

Several mathematical models of CPGs have been proposed to capture the dynamic properties of biological CPGs, such as robustness and flexibility [18,19,20,21,22,23,24]. Additional research has delved deeper into the constituents of CPGs that encompass several rhythmic patterns. These investigations have revealed that CPGs can be flexibly switched between multiple rhythms based on prevailing inputs, which steer the CPG towards one of these rhythms. These models can be classified into phenomenologically and biophysically realistic approaches. Recently, it has been demonstrated that a single CPG can generate multiple rhythms, leading to a multiplicity of emergent rhythmic patterns [25,26,27,28]. Several computational studies have focused on the attraction wells of small neural networks, which draw inspiration from biological circuits such as the crustacean stomatogastric ganglion (STG) pyloric CPG [29], as well as the Tritonia swim CPG [30]. These studies have explored the dynamics of three- and four-cell neural networks coupled by chemical (excitatory, inhibitory, or both types of synapses) as well as electrical synapses, revealing the ability of these networks to produce multiple rhythmic patterns [31,32,33,34,35,36]. One of the aims of this research trend was to investigate the impact of synaptic coupling variations on network capacity to generate distinct bursting patterns of activity.

In many CPGs, particularly spatially symmetric ones, such as the swim CPGs of two sea slug species, *Melibe leonina* and *Dendronotus iris* [9], there is a common building block called a half-center oscillator (HCO) [37]. The HCO, comprising two mutually inhibiting neurons or neuron populations, produces anti-phase bursting, which is its key feature. Our long-term goal has been to develop, as accurately as possible, the biologically plausible models of these swim CPGs to understand how they support their rhythm-generating dynamics, maintain stability (including structural), and remain flexible yet resilient to perturbations.

A few mathematical models of the *Melibe* swim CPG were developed based on its biological counterpart to explore the dynamical foundations of the emergent network-level bursting HCO rhythm [38,39,40]. The circuitry of the CPG was found to be complex (see Figure 1) despite the relatively simple movement of lateral bending swimming. The mathematical model used here was constructed in a simplified configuration of the *Melibe* swim CPG, as previously developed in Ref. [39]. Figure 2 illustrates the reeducation stages toward the simplified CPG model. To qualify as an HCO demonstrating an emergent bursting rhythm, experimental studies on these two specific circuits have de facto shown that their interneurons must display either solely tonic-spiking activity and quiescent states or a combination of these behaviors when isolated. When reciprocally pair-wised coupled and tuned to produce specific dynamical properties, tonic spiking, and quiescent interneurons create a bursting anti-phase emergent pattern network [40]. Non-endogenous bursting neurons, such as tonic spiking and quiescent interneurons, still require dynamic mechanisms such as synaptic escape and post-inhibitory rebound to function properly [41,42].

The electrophysiological recordings were acquired using Spike2 ver. 8 software at a sampling rate of 3 kHz. Although the voltage traces may appear continuous, they consist of discrete values; see Table 1.

The voltage values (in mV) were time-sampled and represented as a time series, like in Figure 3.

Subsequently, the voltage data were imported into MATLAB and stored as vector arrays for further analysis. Determining the sampling rate used during the electrophysiological recordings is crucial as it is utilized at a time step of 1.05 ms in the ODE integration algorithm, ensuring that the mathematical and biological neurons run synchronously. Aligning the sampling rate with the integration step is a critical step in the analysis. Failure to do so results in a misalignment in the temporal scales of the mathematical and biological time series.

Experimental studies using tetrodotoxin (TTX) have effectively demonstrated that this neuroblocker, as illustrated in Figure 4 when applied to the pedal commissure (where axons cross contralaterally), resulted in the following: (i) the decoupling of interneurons in the *Melibe* swim CPG (as shown in A); (ii) the cessation of normal network bursting (B); and (iii) the revelation that the interneurons remained tonically active with similar spike rates when isolated from each other (C). Furthermore, swim interneurons only exhibited quiescence or tonic-spiking activity in isolation even when injected with hyper- or depolarizing currents. In this study, the biological recordings of the CPG were confined to voltage time series obtained through neuronal electrophysiology experiments (intracellular recordings).

The periodic bursting observed in the *Melibe* swim CPG has been identified as a network-level phenomenon that emerges stably due to nonlinear and reciprocal interactions among its coupled interneurons involved in rhythm generation, as depicted in Figure 1. Based on the TTX experiments, the AP frequency in Si3 (swim inter) neurons was found to be higher than in the Si2 neurons during swim episodes. This observation should be accounted for in the mathematical modeling of the system.

To create mathematical models of biological CPGs, it is necessary to compare mathematical time series to biological recordings. This comparison involves analyzing the voltage output, which is obtained from electrophysiological experiments in biological models and simulations in mathematical models. However, this approach suffers from two drawbacks: it is both time-consuming and imperfect. Although mathematical models may closely resemble biological neurons, there can still be minute differences between the two, such as variations in the shape of APs.

Comparing biological recordings from electrophysiological experiments to their mathematical models is a time-consuming process that involves manual processing. This low rate of comparison highlights the need for a high-throughput and accurate method to streamline model construction. To address these imperfections and reduce time consumption, the development of an error function is necessary to quantify differences between biological and mathematical voltage time series. While various methods have been proposed for comparing neural time series, they fail to provide a quantitative measurement for weighing and comparing separate characteristics [43,44,45]. The ability to quantify separate characteristics is crucial for expanding the scope and flexibility of the error function, thereby accurately weighing relevant qualities in computational studies.

In this study, we compare a number of the so-called *blended* CPG circuits, which incorporate voltage recordings measured in the biological swim CPG interneurons to train and optimize the parameters of mathematical neurons to generate “similar” voltage traces as outcomes. First, we develop a *blended* synaptic current to inject into a mathematical neuron. The key component of this current is the synaptic probability or neurotransmitter release rate or transmission rate as a nonlinear function of the voltage variable (mainly due to the action potential (A) or spike frequency) conserved in biological recordings. Such a synapse is modeled by a single differential equation (or an ODE system) that simulates a possible biophysical activity of a biological synapse to be coupled or unidirectionally integrated into a mathematical model called a math neuron. To synchronize or time-wrap the biological recordings, we execute our mathematical models—both cellular and synaptic—at a fixed time step that matches the sampling rates of the experimental voltage time series. This can be conducted using an Euler and fourth-order Runge–Kutta ODE solver with a constant time step, provided the date sampling rate is high enough. To evaluate the “healthy” parameter space of the error function, we conduct a grid search of the conductance of the inhibitory and excitatory synapse from the biological recordings and create an error parameter space.

In what follows, we will present neurophysiological data, specifically intracellular recordings from the swim *Melibe* CPG interneurons in normal conditions and a curare bath, and what pivot cellular qualities and qualities matter most for our study. Next, we introduce the model of the synapse and how time-varying strength correlates strongly with the spike frequency in presynaptic neurons, biological or mathematical. After that, we introduce the math neuron model and discuss its pivot qualities. The blended CPG networks are introduced to incorporate biological and mathematical neurons and to train the latter. Their outcomes are measured and compared against each other to find the degree of their likeness with the combined toolkit of error or cost functions proposed. At the end, we will discuss our findings and future work.

## 2. Methods

### 2.1. Blended Synapse

To simulate the dynamical properties of the swim CPG, it is necessary to model its intrinsic coupling through synaptic interactions, varying from a weak to a strong drive, depending on the spike frequency in presynaptic neurons, which can nonlinearly affect and modulate the dynamics in the driven postsynaptic neurons. In a blended system, the mathematical synapse must unidirectionally translate the activity of a biological neuron into a mathematical neuron, as well as facilitate interactions among mathematical neurons when coupled. To achieve this, we utilized the features of the synaptic current, represented as follows:(1)Isyn=gsynS(t)(Vpost(t)−Erev).
Here, gsyn is the maximal (constant) conductance of the chemical synapse, Vpost is the membrane potential of the targeted postsynaptic neuron, and Erev is the synaptic reversal potential. In our study, we set the reversal potential to Erevinh=−80 mV for inhibitory chemical synapses and Erevexc=+40 mV for excitatory ones, in line with previous modeling studies [39]. However, experimental evidence suggests that reversal potentials are not uniform across all neurons. For instance, the reversal potential for swim interneuron Si2 was found to be Erevinh=−80 mV, while that for Si3 was Erevinh=−50 mV. Additionally, excitatory synapses in the *Melibe* CPG had a reversal potential of Erevexc=−10 mV.

To translate the voltage Vpre recordings of a biological interneuron into the synaptic probability 0≤S(t)≤1, we utilized the single ODE equation, as follows:(2)dSdt=α(1−S)1+e−k(Vpre(t)−Vth)−βS,
where *k*, α, and β are two positive constants that, loosely speaking, determine how quickly the initiation of an inverted growth of the synaptic probability is, and the rate of its exponential decay due to β. Here, the parameter, Vth, denotes the synaptic threshold. When the voltage, Vpre, in the presynaptic neuron drops below Vth, the synaptic probability, S(t), decays following the linear ODE, S′(t)=−βS. Conversely, as long as Vpre exceeds Vth, the differential equation for S(t) above temporally acts as S′(t)=α(1−S)−βS, which makes S(t) increase toward its possible fixed value S*=α/(α+b)≤1. The value of Vth is set between the AP peak and its hyperpolarized value at some threshold set between −20 and 20 mV, allowing S(t) to oscillate quickly or slowly between [0, 1] depending on the values of α and β calibrated selectively to match the admissible spike frequency range in the presynaptic neurons. Let us reiterate that when the presynaptic neuron fires an AP, the synaptic probability is maximized, for example, during active phases of bursting. In contrast, S(t) decreases when the AP falls below the synaptic threshold (Vpre<Vth); this infers that the neurotransmitter release does not occur, and neuron communication either weakens or becomes non-existent during such episodes, for example, during the quiescent phase of bursting. When *k* is large, the switching from Vpre<Vth to Vpre>Vth and back is nearly instantaneous, rendering the equation a continuous sigmoidal approximation S(t)=1/1+e−k(Vpre(t)−Vth) of the Heaviside step function, which is also termed as a fast threshold modulation (FTM) [46].

The above synaptic Equation (Equation 2) is commonly referred to as an α- or α-β-synapse that describes the first-order kinetics [46] of chemical synaptic signaling. It incorporates the logistic function that modulates the transition from a closed to an open state, represented by α. This equation is based on earlier work that simulated synaptic potentials using the α-function [47]. The transition rates between the open and closed states are governed by two constants, α and β, where α denotes the rate of transition from a closed to an open state and β represents the rate of transition from an open to a closed state. The α-values correspond to the strength of binding of neurotransmitters to post-synaptic receptors, whereas the β-values are influenced by various factors, such as enzymatic inactivation, diffusion, and the presynaptic reuptake of neurotransmitters in the synaptic cleft.

In instances where β≪α, the rate of depletion of neurotransmitters is slow, leading to an accumulation of neurotransmitters even at low AP frequencies. Consequently, the probability of the system being open is higher than closed. For example, in the case of the *Melibe* swim CPG, whose active interneurons demonstrate spike rates varying between 5 and 12 Hz during the swim episodes (see Figure 4), fast synapses can be realistically modeled with the constants set around α=0.05 and β=0.005, while slow synapses are better represented with smaller values, such as α=0.02 and β=0.0002, for example. Lowering α reduces the synaptic speed, necessitating a decrease in β to avoid an abnormally fast increase in the accumulation rate.

A significant characteristic of CPGs in sea slugs is the sigmoidal relationship between the probability of synapses opening, and AP frequency ranging typically between 5 and 12 Hz. Prior research [48] demonstrated the sigmoidal relationship between vesicle release rate and presynaptic calcium concentration, as presented in Figure 5. To replicate the observed biological relationship between the Si2 and contralateral Si3 neurons and configure the mathematical synapse, we reproduced the sigmoidal relationship between the average of the postsynaptic variable and the AP frequency of the presynaptic neuron, as shown in Figure 6. By estimating an appropriate pair of values for α and β (as manifested by Figure 6B), we were able to achieve the dynamic responses of Si3 neurons that are shown in Figure 5B.

In the utilized model, the determination of a synapse as slow or fast is exclusively based on the α and β values, with no other parameters involved. If the value of α significantly exceeds that of β, synaptic probability attains saturation at low frequencies. In other words, the synaptic ion channels open with minimal stimulation and persist in an open state, as demonstrated by the upper characteristic curve (red dots) in Figure 6B. Conversely, when the value of β is approximately equivalent to α, the synapse becomes fast and, thus, necessitates a higher spike frequency in the presynaptic neuron to achieve an expected saturation plateau, if any, in its characteristics. This phenomenon is illustrated by the lower characteristic curve (yellow dots) in Figure 6B. One can see from this figure that the average synaptic probability 〈S〉 builds up nonlinearly with the increase in the interspike frequency in the presynaptic interneuron. This implies the synapse of the synaptic current remains weak at low frequencies and may become fully saturated at frequencies higher than 10 Hz, depending on its calibration.

Inspired and backed up by accurate neurophysiological experiments that detail the correlation between spike frequency variations in presynaptic interneurons and the corresponding responses—known as IPSP/EPSP, abbreviations for inhibitory and excitatory synaptic potentiation, or deviations in voltages observed in the postsynaptic neurons—we carefully calibrate the time constant in the synapse model (Equation 2) to ensure that the synaptic probability S(t) strongly correlates (varies) with the spike frequency in the active presynaptic neurons, both biological and mathematical. This feature is manifested in all simulations involving voltage traces in our blended networks; see Figure 7, Figure 8, Figure 9, Figure 10 and Figure 11, where the voltage traces are aligned with simulated synaptic outcomes to underscore this key property, which happens to be pivotal for the successful and plausible modeling of flexible, non-stiff dynamics in oscillatory networks, like swim CPGs in sea slugs. For example, one can see from Figure 7 that the synaptic parameters α and β are adjusted so that the corresponding time-varying probabilities, S(t), corresponding to the color-matched voltage traces of the interneurons S1/2L and 3R are maximized and minimized to match the fluctuations of the spike frequencies during the bursts, demonstrating some small summation due to spontaneous spike trains during the quiescent phase.

### 2.2. Mathematical Model of the Swim CPG Interneurons

The mathematical model used to simulate and match biological swim interneurons (Sis) in the *Melibe leonina* CPG is a specific adaptation of the original Plant model [49,50,51,52], which was derived using the conductance-based Hodgkin–Huxley (HH) formalism. The Si model was calibrated to demonstrate quiescent or tonic spiking activity only, without generating bursting endogenously or in response to perturbation of external currents. Its key features are a pronounced post-inhibitory rebound and a noticeable spike frequency adaptation in response to hyperpolarizing pulse perturbations, like the behaviors of their biological counterparts. A detailed analysis of the Si model and the pairing of matching synaptic properties to build pairwise rhythm-generating circuits can be found in Ref. [53].

The spike generation mechanism of the SI model, concerning its fast dynamics, resembles that of the original HH model. It consists of fast-inward sodium and calcium currents, II, an outward potassium current, IK, and an ohmic leak current, Ileak, bidirectionally coupled through the voltage, V(t), with a slow TTX-resistant sodium–calcium inward current, IT, and an outward calcium-activated potassium current, IKCa, as summarized in the following equation:(3)CmdVdt=−II−IK−Ih−IKCa−IT−Ileak−ΣIsyn.
Here, the *h*-current, Ih, is another quickly depolarizing current that becomes active when the membrane voltage falls below −50 mV. The inclusion of Ih in the Si model prevents it from over-hyperpolarization. Additionally, the term ΣIsyn represents the flow of various synaptic currents, inhibitory, excitatory, and electric, from other presynaptic neurons, as described by Equation (Equation 3). Detailed descriptions and equations for both fast and slow currents are provided in the Appendix A section below.

We will next elaborate in detail on the slow dynamics of the Si model, which are attributed to the outward calcium-sensitive potassium current, IKCa, and the TTX-resistant calcium current, IT. The first current is typically expressed as follows:(4)IKCa=gKCa[Ca][Ca]+0.5(V−EK),
where gKCa is the conductance of the channel, [Ca] is the intracellular calcium concentration, *V* is the membrane potential, and EK is the potassium equilibrium potential. Meanwhile, the intercellular calcium dynamics, defining the pace of IKCa, obey the following ODE equation:(5)d[Ca]dt=ρKc×(ECa−V+ΔCa)−[Ca],
where ρ is a small scaling factor designed to slow down the rate of change of [Ca], Kc is the calcium equilibrium constant, and ΔCa (mV) is a control parameter that allows for the manipulation of the calcium reversal potential ECa within the range of 80 mV to 140 mV. See the Appendix A below for details.

It is worth noting that the slow dynamics of the Si model are mainly attributed to the calcium-dependent conductance of the IKCa channel and the persistent activation of the TTX-resistant IT channel. These mechanisms, coupled with the intercellular calcium dynamics, result in the gradual spike frequency adaption of the membrane potential, which is the key feature of this calibrated neuron model. Overall, these findings shed light on the physiological relevance of calcium dynamics in shaping the electrophysiological properties of the biological CPG interneurons.

The properties of the second current, represented by IT=gT×(V−EI), are determined by the slow dynamics of the voltage-gating variable governed by the following differential equation:(6)dxdt=x∞(V+Δx)−xτx(V).
Here, the bifurcation parameter Δx (mV) was introduced as a deviation from the voltage level at which the *x*-gating variable becomes half-activated, i.e., x=1/2.

A bifurcation diagram of the model with these parameters is presented in Figure 12. One can see that it is partitioned into three major regions of activity: tonic-spiking, bursting, and hyperpolarized quiescent, and their borderlines in the ΔCa,Δx-parameter plane.

By manipulating two parameters, the slow dynamics of the Si model can be modulated to produce either hyperpolarized quiescent or tonic spiking activity for Δx values below −2.5 mV. The outcome is contingent on the ΔCa parameter. Further information on the dynamical and bifurcation characteristics can be found in a recent study [53]. In this study, a value of the parameter Δx=−3.5 mV was fixed to ensure that the Si model bursts endogenously. The dynamics of the targeted Si2 and Si3 neurons in the *Melibe* swim CPG were calibrated by varying the primary control parameter, ΔCa. The Si model remained inactive or quiescent as long as the value of ΔCa exceeded −30 mV, whereas deeper hyperpolarization of the Si model was observed with increasing ΔCa values. Conversely, the Si model exhibited tonic firing with ΔCa values below the threshold of around −30 mV. To account for the difference in AP frequencies between the Si2- and Si3-neurons, their respective ΔCa-values were set to −44 mV and −54 mV. This ensured that the Si3-neurons exhibited higher spike rates than the Si2-neurons, consistent with TTX-experimental observations (see Figure 4).

### 2.3. Blended System Setup

#### Blended CPG Network

In a conventional computational study, the key factor in the strength and time progression of the synaptic current described by Equation (Equation 1) is the time-varying value of the synaptic probability or neurotransmitter release rate given by Equation (Equation 2) in the presynaptic neuron during its active phase—when its voltage is above the synaptic threshold, i.e., outsourced from another mathematical neuron, so to speak. In our novel computational approach, referred to as “blended”, synaptic coupling (or drive) is evaluated from the voltage data recorded from the biological neurons in neurophysiological experiments. Specifically, at each time step of the integration process, the biological voltage value is directly incorporated from a pre-recorded voltage trace (see Figure 7). The details of this process will be expounded upon in the subsequent section.

Let us first describe the configuration shown in Figure 8(A1). The biological neurons of the *Melibe* swim CPG represent a neural cluster, an HCO, of pair-wise coupled non-endogenously bursting neurons Si1/2, which generate the anti-phase bursting rhythms shown in Panel B1 due to reciprocally inhibitory coupling. This network bursting is slow as the CPG is exposed to the curare bath, which blocks any synaptic feedback originating from the interneuron Si3, and influences the pace and dynamics of the top HCO. In this unique setup, the interneuron 1/2L (R) simultaneously provides a contralaterally excitatory (represented by the triangle) drive to interneuron 3R (L), and an ipsilateral inhibitory (represented by a circle) drive to partition its burst initiation and termination.

The *blended* network replica in Figure 8(A2) comprises two biological neurons (blue circles) and two mathematical neurons (red circles). The mathematical neurons that we will refer to briefly are the adapted Si model, which is set to be a quiescent mode to the right from the tonic-spiking border in the parameter place in Figure 12. The mathematical neurons in this blended setup are meant to receive the same kinds of stimulating drives as the biological ones and, therefore, respond as biological originals, 3L/R. The outcomes of the simulations are demonstrated in Figure 8(B3,B4), showcasing the overlaid voltage traces of the biological neuron (3L), its replacement (ML), the bio-neuron (3R), and the math neuron (MR). A detailed examination of the voltage traces confirms they passed an eyeball test, with close spike frequency distributions demonstrated by the traces on both bio- and mathematical neurons, close phases of bursting and quiescent models, even characteristic spontaneous spikes seen in both, and rather good agreement between the corresponding synaptic probability levels.

The configurations shown in Figure 9 demonstrate the normal swim pattern generated by the *Melibe* CPG in the control as well as the blended network. The main difference between this swim pattern and that in the curare bath is its short period that fluctuates within a 4–6 s range. This is due to the reciprocal inhibitory feedback from the interneurons 3L/R; see the circuitry in Figure 2B that is absent in the curare case. The other distinction is the presence of the reciprocal inhibitory coupling that is supposed to make anti-phase bursting between the neurons 3L and 3R more pronounced and, therefore, the emergent oscillations in the whole circuit more stable.

As above, the *blended* network in Figure 9(A2) includes two biological neurons (blue circles) that provide the same characteristic excitatory and inhibitory drives to the mathematical neurons (red circles). In addition, the mathematical neurons are coupled reciprocally by the inhibitory synapses, just like in the bio-CPG. The results of the simulations are shown in Figure 9(B3,B4), representing the overlaid voltage traces of the bio-neurons (3L/R) and the math neuron (ML/R), respectively. By varying, or more accurately, by minimizing the values of the corresponding maximal conductance, ge and gi, in the blended excretory and inhibitory synapses given by Equation (Equation 1), we can regulate their strength to optimize the voltage traces of the mathematical neurons to match the original biological recordings. The typical characteristics include the spike frequencies on the driven mathematical neurons, burst initiations and terminations, their phases, as well as the possible synaptic probabilities to match those of biological neurons. As before, one can attest that the close examination of the voltage traces confirms that it passed an eyeball test, with a close spike frequency distribution demonstrated by the traces of both bio- and mathematical neurons, close phases of bursting and quiescent models, as well as a good agreement between the corresponding synaptic probability levels.

In the following, we propose and discuss the cons and pros of our empirical approach, which aids in substituting or enhancing/complementing the qualitative eyeball test with more quantitative computational algorithms to measure the degree of “likeness” between biological recordings and simulated outcomes of the corresponding mathematical models, and test it on several blended configurations, which we have already discussed, and will present below near the end of the paper.

## 3. Error Functions

In this section, we discuss how the combined error function (CEF) is derived by integrating five distinct sub-functions designed to quantify specific aspects of neural bursting activity generated by the CPG neurons and their math models. These functions were empirically combined using a weighted sum approach, as outlined in Equation (Equation 13) below, in order to best allocate appropriate emphasis to the relevant features. These include a commonly used spike count per burst, as well as measures of quiescent phase proximity and active phase initiation and cessation. In order to avoid the possibility of data peeking, these qualities, with the exception of the spike count, were indirectly quantified.

The first function is the absolute value of the spike count difference, denoted as Δspikes, defined by Equation (Equation 7), which measures the difference between the spike counts of the original biological counterpart and the corresponding mathematical neuron as follows:(7)Δspikes=mspikebio−mspikemod,
where mspikebio is the number of spikes in the biological recordings and mspikemod is the spike count of the mathematical time series of the same length. The spike count algorithm is determined by using the method described in Appendix A, specifically by Equation (Equation 26).

The second function, denoted as |S| in Equation (Equation 8) below, is the synaptic distance, which quantifies the error in the synaptic probability distribution between the neurons. To calculate |S|, we evaluate the difference between the biological and mathematical synaptic probabilities, and take the square root of the sum of squares (SRSS) of the differences. We use the synaptic probability instead of the voltage as it is a smoothing function of the same neuron voltage while preserving the information that informs about the AP frequency. Note that the voltage amplitude in biological data can fluctuate substantially, especially in the case of intracellular recordings where the electrode is poked in the soma of the same targeted cell. The synaptic probability is given by the following equation:(8)∥S∥2=∑i=1NSibio−Simod2,
where Sbio and Smod denote the synaptic probability functions of the biological recordings and the mathematical neuron, respectively (see Equation (Equation 27) in the Appendix A for more information).

The third function is the voltage distance of the moving average, denoted as |V^| in the following equation:(9)∥V^∥=∑i=1NV^ibio−V^imod2.
We calculate |V^| using the SRSS of the differences between the voltage moving averages of the biological and mathematical time series. The voltage moving average of the mathematical time series, V^mod, and biological recordings, V^bio, are as follows:(10)V^=1k∑i=N−k+1NVi,
where V^ is the moving average, defined as the average of the *k* last entries of the voltage time series of the *N*-length.

The fourth function quantifies the similarity of compared patterns as the variance of the difference in the voltage’s moving average, s2, given by Equation (Equation 11). A value of zero indicates that the pattern is perfectly aligned between the two moving averages along the time axis. The function is defined as follows:(11)s2=1N−1∑i=1N|V^ibio−V^imod|−μ2,
where μ is the mean of |V^bio−V^mod|.

The fifth function is the sigmoid of the distance in the enveloped voltage time series, denoted as |E|, and is given by Equation (Equation 12). This function normalizes the error values by calculating the upper and lower envelopes of a voltage time series. It then determines the SRSS from the upper and lower envelopes of another voltage time series. Specifically, it is defined by the following equations:(12)∥E∥=fσ(∥L∥+∥U∥),where∥L∥2=∑i=1NLibio−Limod2,∥U∥2=∑i=1NUibio−Uimod2,fσ(x)=1/1+e−a(x−c),
where Lbio and Lmod are the lower envelopes and Ubio and Umod are the upper envelopes of the biological recordings and the mathematical voltage time series, respectively. The sigmoid function is used for normalization with some positive, and empirically defined constants, *a* and *c*. The envelope is further discussed in Appendix A below. In summary, the fourth and fifth functions are essential for quantifying the pattern similarity and normalizing the error values, respectively. These functions are prepared for combinations by rescaling their output to a domain defined on a unit interval [0,1].

## 4. Results

The synaptic strength in the blended synapses is regulated by the corresponding maximal conductance gex and gin from Equation (Equation 1), correspondingly. We use these two parameters to optimize the outcomes of the mathematical neurons to match the original biological recordings. We note that, on their own, the individual functions introduced above can only provide a limited understanding or give biased insights to compare and quantify distinct types of neuronal rhythmic activity. So, while the synaptic distance function |S| given by Equation (Equation 8) measures the error in the active phase of a neuron, it fails to differentiate between inactive phases, as all of them have a synaptic probability of zero. Consequently, this function alone is not a proper characteristic of the neuronal activity. Additionally, a positive relationship is observed between the excitatory synaptic conductance and error, as demonstrated in Figure 13(A1). The minimum error time series (MET) for the |S| function is found to occur when ge=0 and gi=0.0105, as can be observed from Figure 13(A2). However, the synaptic distance function does not adequately weigh the importance of the AP frequency either.

The spike difference function, Δspikes, defined by Equation (Equation 7), also has its own limitations, as it demonstrates two local minima shown in Figure 13(B1), and does not differentiate well between active and inactive phases of bursting. While the MET for this function qualitatively matches the data well, as manifested by the shape of the function revealed in Figure 13(B2), the value of gi has little effect on the error space in the Δspikes error space, as one can observe from Figure 13(B1). For instance, with gi=0,ge=0.395, the voltage time series of the math neuron would continue spiking during the inactive phase in the biological trace, as illustrated in Figure 14(A2). Overall, these findings highlight the importance of considering multiple functions for a better characterization of neuronal rhythmic activity.

The quantitative evaluation of rhythmic activity by neurons is also crucial for a deeper understanding of the neuronal mechanisms underlying various physiological processes. Existing error functions, such as |S| and Δspikes, fail to capture the qualitative characteristics of a neuron’s rhythmic activity. Another function, |V^|, is based on the moving average voltage distance as it can distinguish between inactive and active phases of bursting in voltage traces but treats both phases as equally valuable contributors to the rhythmic activity of the neuron. As such, it may fail to account for the higher importance of an active phase, which contains a wealth of pivotal data compared to interburst intervals, which can be only differentiated by their durations and the baseline, representing the voltage equilibrium level. Similarly, the difference variance function, s2, is more discerning than |V^|, but may still allow the neuron to continue spiking during its inactive phases. Both |V^| and s2 measure the moving average time series, which cannot differentiate whether the neuron is firing AP, only that it is above the spike threshold.

The distance in the enveloped voltage time series, |E|, has no apparent preference for either the active or inactive phase. However, there is a high variance in the error values associated with |E|, and the most effective target for the MET involves overexposure to excitatory conductance, as evidenced by the rising active phase minimums. Furthermore, |E| cannot account for AP frequency adaptation due to its inability to measure AP frequency. In summary, |S| fails to account for the active phase, |V^| and s2 overlook the inactive phase, and function Δspikes cannot detect it (as it was not designed for this purpose). Additionally, function |E| has a high variance and cannot well differentiate between AP frequencies.

However, we argue below that the combination of the five functions can effectively compensate for their intrinsic faults by using a proportioned-weighted sum WS, defined as follows:(13)WS=w1Δspikes+w2∥S∥+w3∥V^∥ + w4s2+w5∥E∥,
with the respective weights of the functions, denoted as wi, are chosen manually as follows: w1=0.10, w2=0.58, w3=0.10, w4=0.19, and w5=0.03, where *i* denotes the function index. The selection of these weights is subject to constraints ensuring that the error space meets certain desired criteria, namely, the absence of AP firing during the quiescent phases, synchronization of burst onset and termination, and a quiescent phase that does not deviate significantly from the biological reference. Thus, the defined weighted sum WS given by Equation (Equation 13) allows for mutual compensation of the individual functions, as demonstrated in Figure 15.

A combined error function (CEF) was constructed using these weights. To ensure the generalizability of CEF, a regularization term is usually added to prevent overfitting. However, in this study, such a term is not included as it would require a comparison of the fit to the goal voltage recording, which raises the issue of comparing the raw voltage time series.

Despite the limitation of not being able to achieve adequate objective regularization through the error function, we explore the error space of three different endogenous states of the biological and mathematical neurons and their METs. These states include tonic spiking to the left of the borderline state around ΔCa=−30 mV, and quiescent neurons to the right of the parameter space shown in Figure 12. We found that the CEF was able to determine unique gi and ge values (indicated by *) for each state with a minimum error, as depicted in Figure 15.

To further evaluate the performance of the CEF, we blended biological recordings with the mathematical neurons in a specific order. Specifically, we blended the same recordings with tonic-spiking mathematical neurons, then borderline neurons, and finally with quiescent neurons. The MET for the blended system using tonic spiking mathematical neurons was minimized at gi=0.01579 and ge=0.0316. Similarly, the blending with borderline neurons had a MET minimized at gi=0.0118 and ge=0.0316, while the blending with quiescent neurons had a MET minimized at gi=0.0066 and ge=0.0395. Notably, each of these different gi and ge coordinates had similar METs, and no discernible difference was observed among them, as one can observe from Figure 15B).

However, we observed a difference in the error space between each blending with the different mathematical neurons. Specifically, the minimum error decreased as inhibitory conductance increased, corresponding to a shift from more intrinsically active states (i.e., from tonic spiking to borderline and eventually to a quiescent neuron). This finding was consistent with our expectation since the stronger the mathematical neuron is quiescent intrinsically, the less inhibition is needed to suppress its firing during the naturally inactive phases.

One additional verification step was performed to test the effectiveness of the CEF in the context of biological recordings. Specifically, the CEF was blended with normal swimming biological recordings, where the CPG was not exposed to curare. The configuration of the biological CPG in the absence of curare featured a pair of mutually inhibitory synapses between the biological neurons, Si3 (see Figure 9A), which were absent in the simplified CPG model depicted in Figure 2B). To incorporate this feature into the CEF, two corresponding synapses were added between the mathematical neurons to recreate the reciprocal inhibition while keeping the conductance constant, to avoid introducing an additional parameter dimension. The resulting MET for this configuration is shown in Figure 9B. While the MET detects well some types of rhythmic activity similar to the biological recordings, such as overlapping bursts, the AP bursting frequency in the mathematical neurons is not as high as in the bio-recordings. Nevertheless, considering that only two parameters were optimized, the MET shows good agreement with the normal biological recordings.

In the CEF error space of the blended system, employing normal biological recordings resulted in the identification of the MET for normal swimming, as shown in Figure 16. Comparable to the blended system utilizing curare voltage time series and configuration, the optimal fit time series showed a decrease along the inhibitory conductance (gi-axis) and an increase along the excitatory conductance (ge-axis), as the mathematical neurons were transformed step-by-step from endogenous tonic-spiking in panels A1 and A2 in Figure 16, to borderline activity in panels B1 and B2 in Figure 16), to the hyperpolarized quiescent state in Figure 16C1,C2. However, the inhibitory and excitatory conductance values for the normal blended system differed from those of the curare blended system, which is discussed in the subsequent section, and is evident from the dissimilar conductance values indicated in the captions of Figure 15 and Figure 16.

The error value obtained from the analysis of the slow bursting in a curare bath illustrated in Figure 8 is significantly low at 0.073, surpassing even the optimal fits identified through bi-parameter sweeps. This can be attributed to the limited number of parameters engaged in the aforementioned methods, which are restricted solely to the synaptic coupling parameters. Conversely, the hand-tuned blended system includes the tuning of fast subsystem parameters, playing a crucial role in the firing of APs and other related functions. This finding serves as an excellent example of how a meticulously calibrated blended system can yield an accurate fit for the CEF. However, upon a careful examination of the presented Table 2, Table 3, Table 4 and Table 5, a discrepancy in the weighted error value column of the mathematical neuron is observed. The synaptic distance emerges as the primary contributor to the error, while the envelope yields the least error. This indicates that the voltage of the mathematical neuron is not relatively shifted, but rather the spiking frequency is inconsistent with that of the biological recording.

The analysis demonstrates that the error value associated with results concerning the blended network for the normal swim depicted in Figure 9 is around 0.15 (0.1498), as determined by the CEF. This value falls within the acceptable range of a good fit, as per the predetermined threshold set at value 2.2. Further examination of the results presented in the accompanying Table 2, Table 3, Table 4 and Table 5 reveals that the mathematical neuron exhibits a considerable error in the weighted error value column. Notably, the contribution of the synaptic distance to the overall error is significant, while the variance in the moving average difference is comparatively minor. This observation suggests that there is a moderate level of consistency in the phase alignment between the two models. However, the voltage of the mathematical neuron is lower than that of the biological recording.

**Table 2 brainsci-14-00468-t002:** Error values for Figure 8.

Error Function	Error Value	Weighted Error Value
CEF	0.073	
Synaptic	0.0861	0.0497
Spikes	0.1051	0.0101
Volt. MA	0.0981	0.0094
Variance diff.	0.0175	0.0034
Envelope	0.0115	0.0004

**Table 3 brainsci-14-00468-t003:** Error values for Figure 9.

Error Function	Error Value	Weighted Error Value
CEF	0.1498	
Synaptic	0.1014	0.0585
Spikes	0.5008	0.0482
Volt. MA	0.0223	0.0021
Variance diff.	0.2028	0.0634
Envelope	0.0036	0.0002

**Table 4 brainsci-14-00468-t004:** Error values for Figure 17.

Error Function	Error Value	Weighted Error Value
CEF	0.0715	
Synaptic	0.0423	0.0244
Spikes	0.2336	0.0225
Volt. MA	0.1616	0.0155
Variance diff.	0.0418	0.008
Envelope	0.027	0.001

**Table 5 brainsci-14-00468-t005:** Error values for Figure 18.

Error Function	Error Value	Weighted Error Value
CEF	0.0727	
Synaptic	0.0551	0.0318
Spikes	0.2172	0.0209
Volt. MA	0.0589	0.0057
Variance diff.	0.0703	0.0135
Envelope	0.0219	0.0008

The error value for Figure 17 is 0.0715, which is comparable to that of the hand-tuned curare case. Upon inspection, APs in the figure appear thicker than in the corresponding biological recordings (see Figure 19). This discrepancy suggests that the CEF may inadequately represent the AP shape. This observation is noteworthy, as the curare swim case does not exhibit thicker APs despite similar CEF values. Analysis of the four Table 2, Table 3, Table 4 and Table 5 indicates that synaptic distance is the primary error source, with the spike count error also in adequate proximity, while the envelope yields the least error. These findings imply that spike frequency is inconsistent with the biological recording.

Figure 18 demonstrates an error value of 0.0727, which is comparable to the two preceding cases. Upon zooming in, it becomes apparent that the APs in the mathematical neurons are thicker in shape than those in their biological counterparts (see Figure 19). The analysis of the table indicates that, as before, the synaptic distance is the primary error source, while the envelope has the least error and a higher variance difference than the previous case. These results suggest that the AP characteristics significantly contribute to the overall error, potentially due to inconsistencies in spike frequency. Notably, the voltage of the mathematical neuron is not relatively shifted, but the phase alignment is suboptimal compared to previous cases, as evidenced by the higher variance difference.

## 5. Discussion

We presented a novel type approach, denoted as *blended* systems, where pre-recorded neurophysiological data and intra-cellular voltage recordings from swim CPG interneurons of the sea slugs *Melibe leonina* and *Dendronotus iris* are employed to train and optimize biologically plausible mathematical neurons and synapses. These systems have the potential to offer significant advantages in computational neuroscience research, particularly in the field of brain–computer interfaces (BCIs). While BCIs also interact with biological neurons, they differ in that they connect to living neurons rather than pre-recorded ones. The most promising application of BCIs is the restoration of vision, hearing, and locomotion. However, existing algorithms for BCIs do not fully capture—both qualitatively and quantitatively—the observed (voltage) rhythmic activity of biological neurons.

To address this issue, we developed a set of error functions to quantify the closeness or likeness as well as differences in neural activities between biological recordings and mathematical time series. Our results indicate that a weighted summation of five distinct functions (termed the CEF) yields a robust functionality that can distinguish well between voltage time series. When evaluated individually, these functions were insufficient to qualitatively mimic the target biological recording. We hypothesize that the CEF should (ideally) yield a perfect match when its value is zero.

Moreover, due to the convex nature of the error functions in the parameter space of neural systems, it can function as a cost function for artificial neural networks (ANNs) to train them to qualitatively mimic the voltage time series of a mathematical network using limited input parameters. We also utilized the error function to investigate the similarity between the outcomes of a mathematical CPG and the pre-recorded biological CPGs.

In the preceding section, it was observed that the conductance parameters for the METs differed between the normal biological CPG and the curare-bathed system. This unexpected difference can be attributed to the long inactive phases in the bursting of the curare *blended* system, which allows for more forgiving time series transitions between inactive and active phases. Therefore, weaker excitatory and inhibitory conductance can still produce expected results, as evidenced by the conductance parameters of the normal *blended* system, which can be extrapolated to the curare *blended* system.

Additionally, voltage recordings from *Dendronotus iris* were utilized to construct two distinct *blended* systems, each capable of producing an HCO rhythm in the mathematical neurons (refer to Figure 10 and Figure 11). These systems serve as examples of the versatility of the *blended* approach in generating various types of CPGs, drivers, and pacemakers. In Figure 10, the *blended* system emulates the behavior of Si1 neurons in the *Dendronotus* swimming CPG, acting as a driver neuron and inducing anti-phase firing of other neurons in the CPG. Figure 11 presents an alternative hypothetical configuration of a *blended* system that produces an HCO rhythm, where the Si3 voltage recording excites a quiescent mathematical neuron (M1) and inhibits a tonic spiking mathematical neuron (M2), thus creating a typical pacemaker CPG.

The error function proposed in this paper can enhance the optimization of mathematical neuron parameters and aid developers of circuits that interface with biological neurons. However, the computational expense of the CEF may limit its utility. Therefore, further innovation is necessary to devise an error function that can advance computational neuroscience research.

## Figures and Tables

**Figure 1 brainsci-14-00468-f001:**
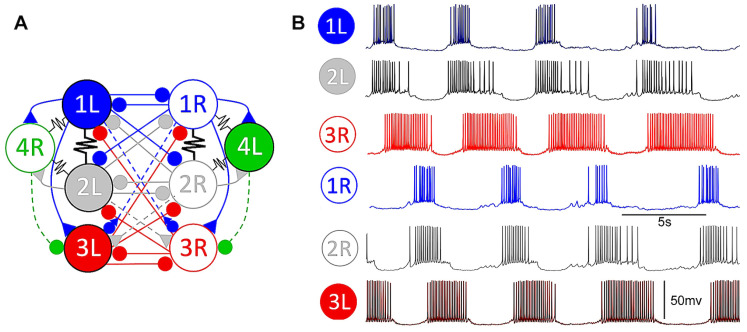
Panel (**A**) demonstrates an intricate circuitry of eight identified interneurons that comprise the center pattern generator (CPG) regulating swimming behavior in the sea slug, *Melibe leonina*. This CPG is comprised of two opposing neuronal populations of several half-center oscillators (HCOs), each consisting of pair-wise inhibitory-coupled right (R) and left (L) swim interneurons (sIs). The synapses here are represented by dashed/solid lines with symbols △, ▽, and •, which indicate slow-/fast-activated synapses, excitatory and inhibitory, respectively, whereas jagged lines indicate electrical synapses. Panel (**B**) demonstrates the synchronized intercellular voltage recordings of the six CPG interneurons during a swimming episode in the control case, revealing the network bursting pattern with fixed phases. Interneurons 1L, 2L, and 3R display specific burst periods of approximately 5 s (in normal swim) and with anti-phase (0.5) phase lags between corresponding contralateral interneurons.

**Figure 2 brainsci-14-00468-f002:**
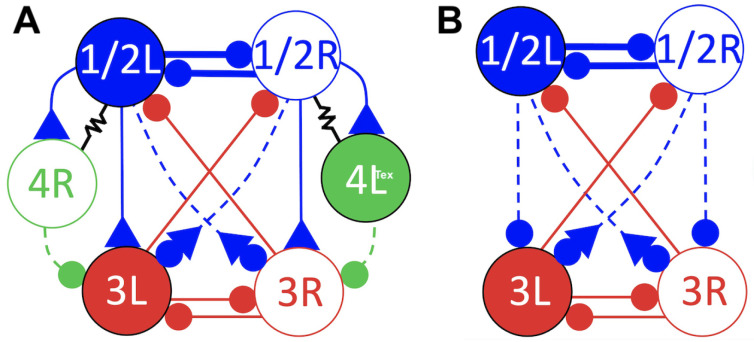
The CPG circuitry in panel (**A**) represents a first-step reduction in the number of ipsilateral interneurons, where interneurons 1L/R and 2L/R, introduced in Figure 1A, are combined into a single cluster due to strong gap junctions connecting them on each side, denoted by 1/2L and 1/2R, respectively, with close bursting phase episodes. The CPG configuration in panel (**B**) is a further reduction, achieved by bypassing the intermediate interneurons 4L and R, following the bursting phases generated by the combined 1/2L and 1/2R due to the feed-forward drive through slightly delayed excitatory synapses. The synapses denoted by dashed/solid lines represent slowly/quickly activated synapses, respectively, while jagged lines denote gap junctions (electrical synapses).

**Figure 3 brainsci-14-00468-f003:**
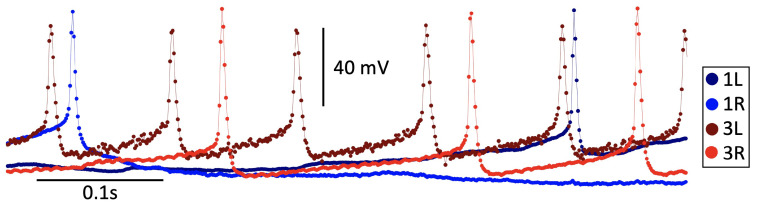
An episode of simultaneous electrophysiological voltage recordings of four CPG interneurons at a sampling rate of 1.05 ms using Spike2 ver. 8 software. To train and optimize the suggested neuron models, we further synchronize these recordings with mathematical simulations using the Euler ODE solver with an equal time step of 1.05 ms. One can observe that relying on time-averaging over voltage traces was hardly feasible to detect and evaluate variations in neural activities due to sparse representations of fast spikes with limited sampling points compared to heavily populated longer inter-spike intervals dominating the recordings.

**Figure 4 brainsci-14-00468-f004:**
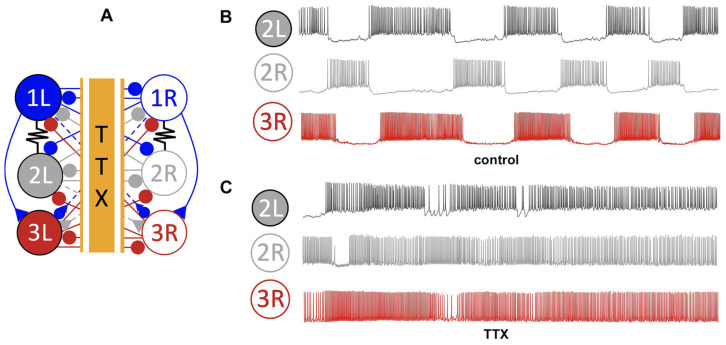
Application of a tetrodotoxin (TTX) neurotoxin on the pedal commissure of the axons crossing contralaterally (i) decouples the opposing interneurons of the *Melibe* swim CPG circuit, as shown in panel (**A**); which results in (ii) ceasing the normal network bursting (the three control traces in panel (**B**)), and (iii) reveals that the interneurons remain tonically active (traces in panel (**C**)) with similar spike rates when become isolated from each other.

**Figure 5 brainsci-14-00468-f005:**
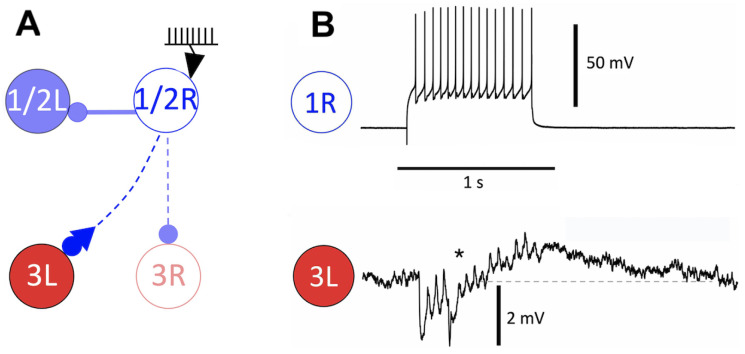
Experimental evidence of a *biphasic* synaptic response in the *Melibe* swim CPG. (**A**) Experimental setup: a spike train is injected into the quiescent presynaptic interneuron 1R to stimulate a burst of action potentials, to examine postsynaptic potentiation in the interneuron 3L coupled with a biphasic synapse ending with a fast •-inhibitory terminal and an adjacent slow ▲-excitatory terminal. (**B**) Stimulation of the interneuron 1R with a train of short depolarized pulses at 15 Hz triggers a biphasic synaptic response in interneuron 3L, consisting of an initial fast hyperpolarization followed by a slow depolarization. The initiation phase of the depolarization is marked by an asterisk in the time progression. The slow depolarization persists until the end of the stimulation, followed by a slow relaxation phase back to the steady state voltage level.

**Figure 6 brainsci-14-00468-f006:**
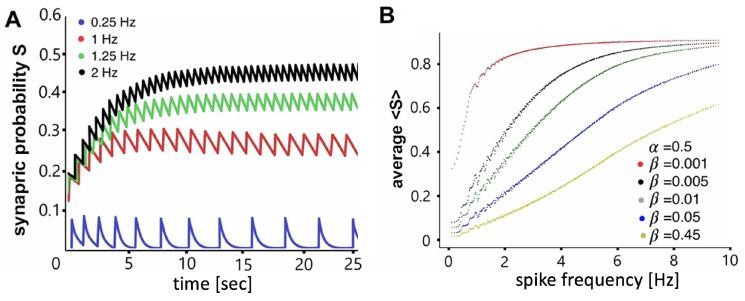
The time-scale calibration of the α-synapse model (Equation 2) to pair it and match it with the range of spike rates observed in the swim CPG interneurons. Panel (**A**) displays a significant buildup of synaptic probability S(t) in a given α synapse as the spike frequency in the presynaptic neuron increases. Panel (**B**) shows the nonlinear dependence of the average synaptic probability, 〈S〉, in the α-synapse model (Equation 2) with fast activation due to a fixed large time constant, α=0.5, and decreasing time constant, β, from 0.5 in the fast synapse to 0.001 in the slow synapse, on the spike frequency in the presynaptic neuron.

**Figure 7 brainsci-14-00468-f007:**
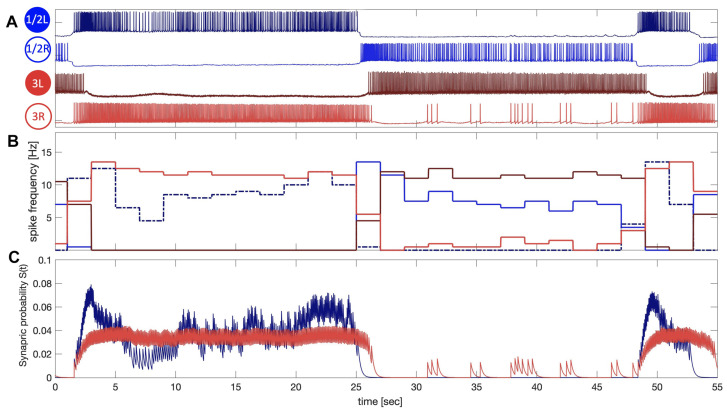
Panel (**A**) demonstrates a long, 12–14 s, bursting alternation pattern between the leading interneurons 1/2L and 1/2R and the following post-synaptic interneurons 3L/R of the *Melibe leonina* swim CPG in a *curare* bath (see Figure 8(A1) below) blocking chemical synapses of 3L/R. (**B**) Spike frequency variations (matching colors) of all four cells are plotted across time, revealing higher spike rates in the postsynaptic interneurons 3L and 3R, driven by expiatory synapses originating in interneurons 1/2R and 1/2L. (**C**) Simulated neurotransmitter release probabilities S(t) throughout a “blended” α-synapse model (Equation 2) demonstrate a strong correlation with specific time-varying spike rates in the voltage recordings of biological neurons 1/2R and 3R shown in Panel (**A**).

**Figure 8 brainsci-14-00468-f008:**
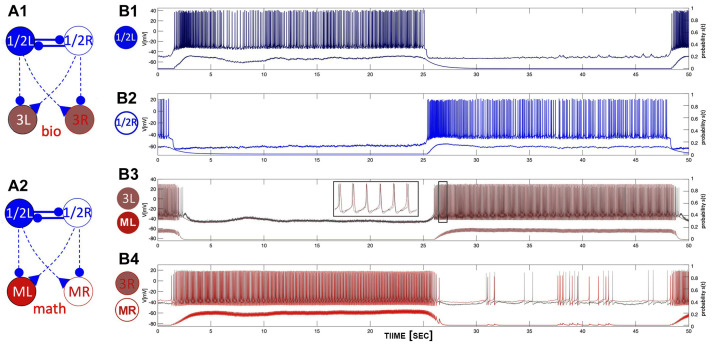
The 4-cell swim CPG in a curare bath (panel (**A1**)) designed to block outgoing synapses of both interneurons 3L and 3R, and a blended network (panel (**A2**)), where biological bursters 1/2R and 1/2L provide (contralaterally) a simulated excitatory drive and (ipsilaterally) a simulated inhibitory drive, resp., onto the mathematical replicas ML/MR of interneurons 3L and 3R. Panels (**B1**,**B2**): slow 12–14 s bursting generated by an HCO due to strong inhibitory reciprocation between interneurons 1/2L and 1/2R in the absence of feedback inhibition from interneurons 3L/R. Panels (**B3**,**B4**): an excitatory drive from the alternating bursters 1/2L and 1/2R causes the counter-lateral quiescent interneurons of both types, biological interneurons 3L and 3R (gray voltage traces), and their mathematical replicas ML and MR (superimposed red traces) to follow, while the bilateral inhibition cuts them off into even spike trains. Here, the traces are superimposed with the corresponding synaptic probabilities emulated through Equation (Equation 2) to model excitatory and inhibitory currents (by Equation (Equation 1)) injected in the mathematical neurons (in red) of the blended circuitry in panel (**A2**).

**Figure 9 brainsci-14-00468-f009:**
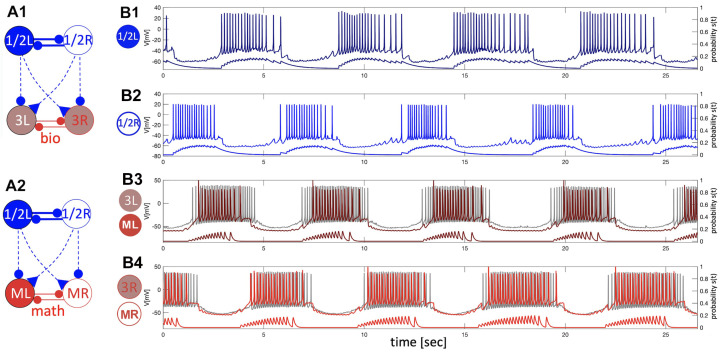
Panels (**A1**,**A2**) show the 4-cell swim CPG in a normal swim case in saline (**A1**), and a blended network (**A2**), both consisting of two biological interneurons, 1/2R and 1/2L, which project an excitatory drive contralaterally and an inhibitory drive ipsilaterally, respectively, onto the biological interneurons, 3L/R, and the mathematical models ML/MR (**A2**). Panels (**B1**,**B2**) depict voltage recordings of the *Melibe* swim CPG interneurons 1L and 1R bursting in anti-phase, which are superimposed below with the aligned simulated traces of neurotransmitter release probabilities S(t) for each neuron. Panels (**B3**,**B4**) demonstrate voltage recordings (in gray) of the biological interneurons 3L and 3R, overlapping with simulated voltage traces (in red) of their mathematical models ML and MR at ΔCa=−30 mV, alongside the bottom traces of neurotransmitter release probabilities.

**Figure 10 brainsci-14-00468-f010:**
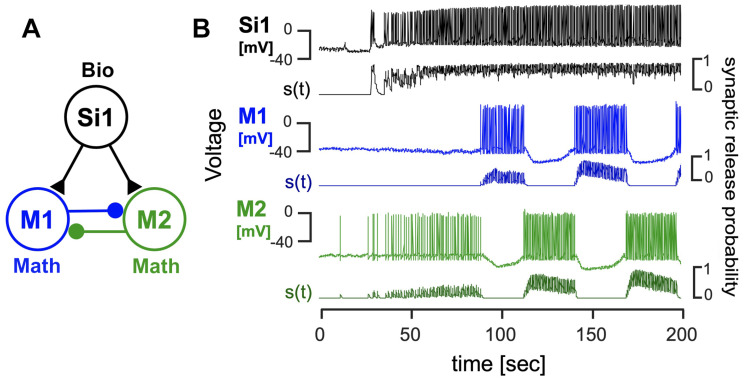
(**A**) A *blended* network where the voltage recordings of the *Dendronotus* swim CPG neuron Si1 are used to activate and induce an alternating bursting rhythm in two quiescent mathematical neurons, M1 (blue) and M2 (green). The excitatory and inhibitory mathematical synapses are indicated by ▲ and •, resp. (**B**) Color-matched voltage time series of the pre-recorded tonically spiking *Dendronotus* interneuron Si1 alongside its simulated synaptic probability, whose excitatory drive injected into two quiescent mathematical neurons makes them burst in alternation.

**Figure 11 brainsci-14-00468-f011:**
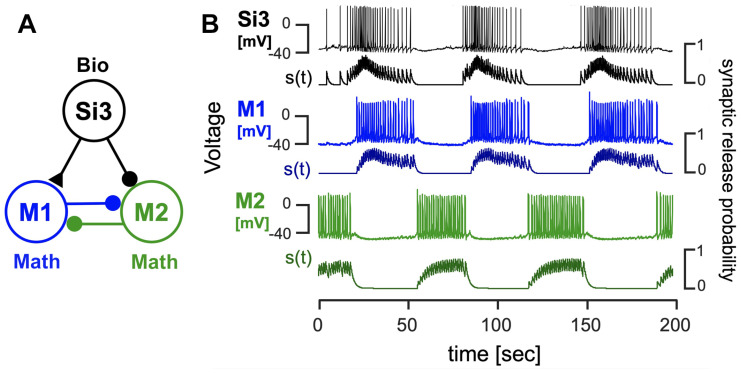
A *blended* 3-cell configuration utilizing voltage recordings of the *Dendronotus* Si3-neuron to induce an alternating bursting rhythm in two mathematical M1/M2 neurons. (**A**) The system schematics include the biological Si3 neuron, which excites (via a synapse with (▲) a quiescent M1 neuron and inhibits (a synapse with •) a tonic-spiking M2 neuron; both M1 and M2 also inhibit each other during their active phases. Panel (**B**) presents a voltage time series of the *Dendronotus* swimming interneuron Si3, and a simulated varying synaptic probability (trace below), along with voltages and synaptic probability traces to model synaptic interactions between the mathematical neurons in alternating bursts.

**Figure 12 brainsci-14-00468-f012:**
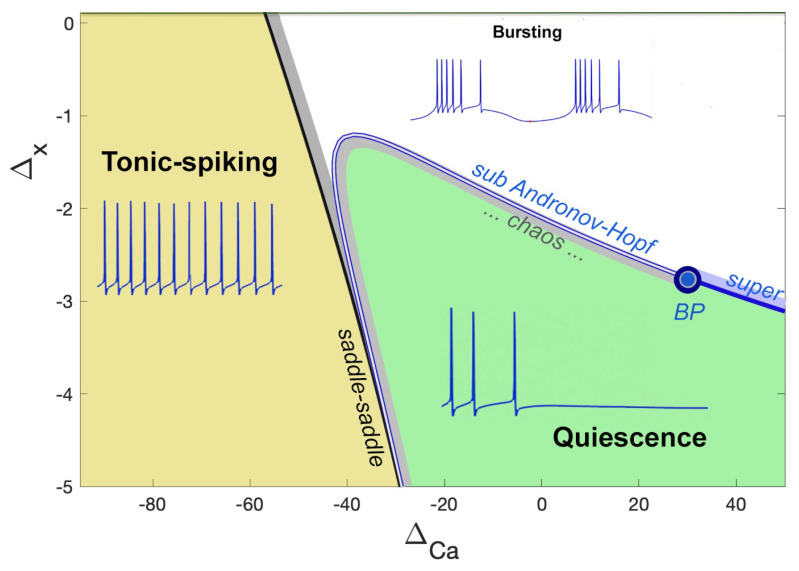
The (ΔCa,Δx)-bifurcation diagram of the Si-neuron model with the three regions corresponding to tonic-spiking, bursting, and quiescent activity. Below the level, Δx=−2.5 mV, the neuron model demonstrates tonic-spiking activity or quiescence only, depending on the ΔCa-value.

**Figure 13 brainsci-14-00468-f013:**
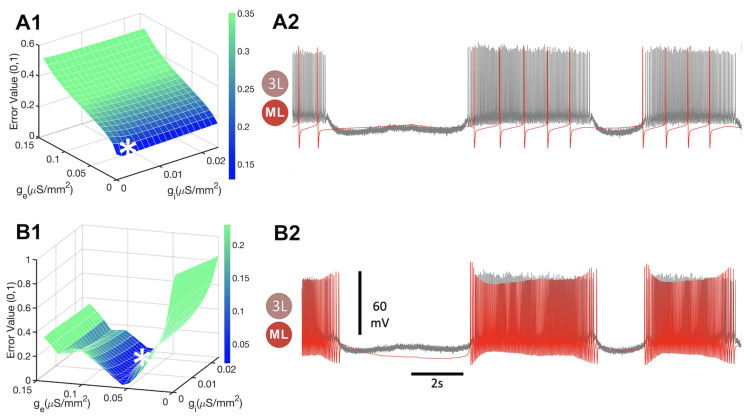
(**A1**,**B1**) Error spaces and minimum error time series (MET) space. The vertical axis represents the rescaled value of the error functions, while the *x*-axis and *y*-axis represent the conductances, gi and ge, of the inhibitory and excitatory synapses, resp. (**A2**,**B2**) Voltage recordings of the bio-neuron (3L) and the math neuron (ML) are compared to find the gi and ge values corresponding to the lowest error value for each function. (**A1**) The synaptic distance error space and (**A2**) synaptic distance MET. The synaptic distance error space evaluated through Equation (Equation 8) at gi=0.0013 and ge=0 (indicated by *) indicates the large mismatch between the spiking frequencies of the mathematical and biological neurons in the curare bath. (**B1**) The spike difference error space elevated through Equation (Equation 7) is minimized at gi=0.0171 and ge=0.0711 (indicated by *), where the spiking frequencies of the mathematical and biological neurons agree well, as seen in panel (**B2**).

**Figure 14 brainsci-14-00468-f014:**
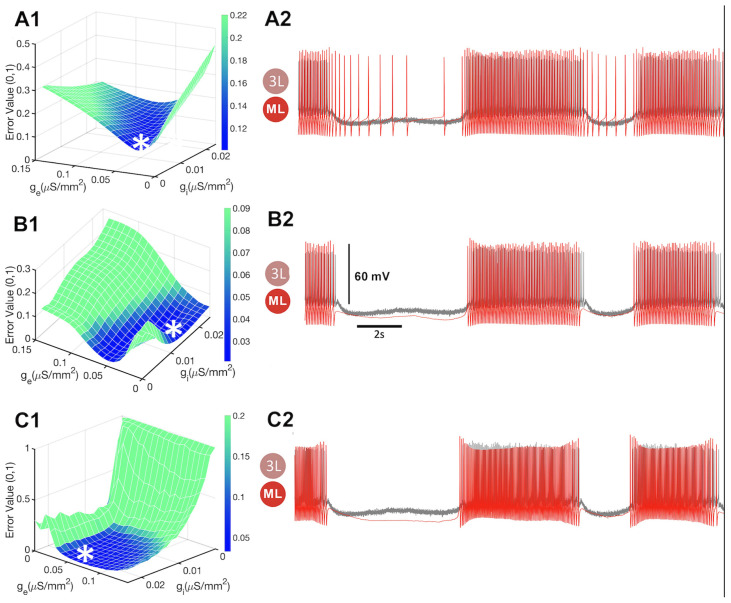
(**A1**,**B1**,**C1**) The *z*-axis represents the rescaled value of the error function, the *x*-axis and *y*-axis represent the conductances, and gi and ge denote the inhibitory and excitatory synapses. (**A1**) Voltage distance error space evaluated using Equation (Equation 9) is minimized at gi=0026, ge=0.0237 (indicated by *). (**A2**) Here, the math neuron keeps spiking during the inactive phase of the bio-neuron. (**B1**) Evaluation of the difference variance error space (DVES) according to Equation (Equation 11). (**B2**) The traces of the math- and bio-neurons do not fully overlap at The local min of MET at gi=0.0145, ge=0.0237 (indicated by *) (**C1**) Envelope distance error function given by Equation (Equation 12). (**C2**) The traces of both neurons match well at the local min of MET at gi=0.0197, ge=0.0553 (indicated by *), although the function can barely detect a spike frequency adaptation in the bio-neuron in the curare bath.

**Figure 15 brainsci-14-00468-f015:**
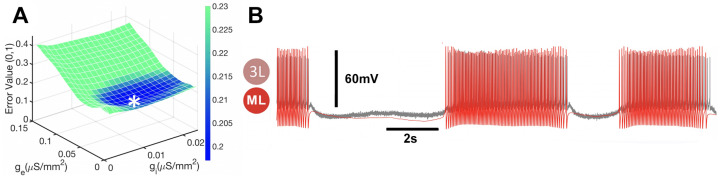
The combined error function (CEF) can effectively adjust to the cellular properties of the mathematical neuron. (**A**) The error space of excitatory and inhibitory synaptic conductance; here * indicates a local minimum foe the optimal conductance values. (**B**) All best-fit time series—whether the Si model exhibits tonic-spiking activity at ΔCa=−40 mV, resides on a borderline at ΔCa=−40 mV, or becomes quiescent at ΔCa=−25 mV—exhibit qualitatively similar voltage time series. The minimum error becomes smaller along the inhibitory conductance axis as the mathematical neuron shifts from a tonic spiking activity with gi=0.01579 and ge=0.0316; borderline with gi=0.0118 and ge=0.0316; to a quiescent neuron with gi=0.0066 and ge=0.0395. Each MET matches the other and those of the biological neurons in the curare bath.

**Figure 16 brainsci-14-00468-f016:**
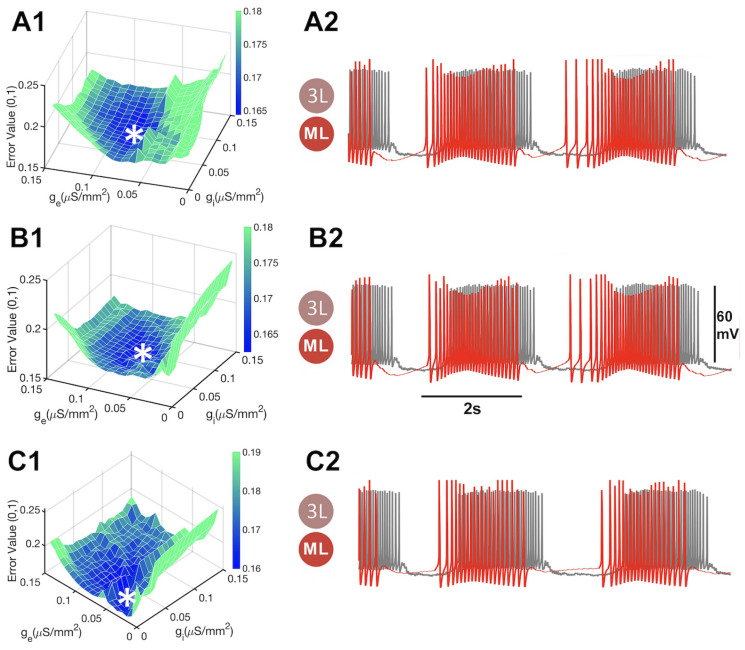
The application of CEF to the normal swim voltage recordings. The minimum error becomes smaller along the gi-axis, similar to the curare case, as the mathematical neuron shifts from (**A1**) tonic spiking activity at gi=0.0711, ge=0.0711 (indicated by *) throughout (**B1**), a borderline at gi=0.0789, ge=0.0789 (indicated by *), and ultimately to (**C1**), a quiescent state at gi=0.0237, ge=0.0474 (indicated by *). (**A2**,**B2**,**C2**) MET-function comparing the mismatch between the neuron (3L) and a mathematical neuron (ML). It is evaluated by varying gi and ge values to determine its lowest error value corresponding to qualitatively similar voltage time series, biological and mathematical. Furthermore, the minimum error becomes larger along the ge-axis.

**Figure 17 brainsci-14-00468-f017:**
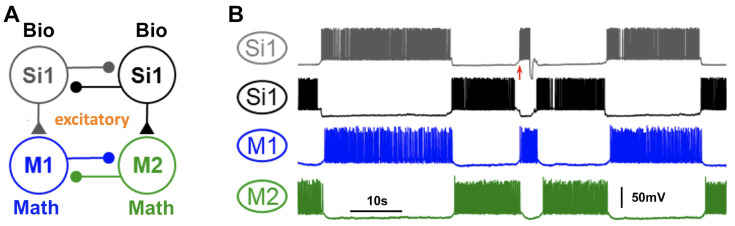
Another *blended* 4-cell network to train the mathematical neurons, M1/2, using the voltage recordings of the biological *Melibe* Si1 interneurons. (**A**) The *blended* circuit where biological Si1 interneurons project excitatory drive (▲) into quiescent mathematical neurons, which are also inter-coupled reciprocally by inhibitory synapses denoted by •. (**B**) Color-matched voltage traces of the *Melibe* interneurons, M1 (blue) and M2 (green), are shown alongside the voltage time series of the two mathematical neurons. The red arrow represents perturbation due to a short depolarized current pulse flipping unsuccessfully in the middle of a long burst in the biological HCO in a curare bath.

**Figure 18 brainsci-14-00468-f018:**
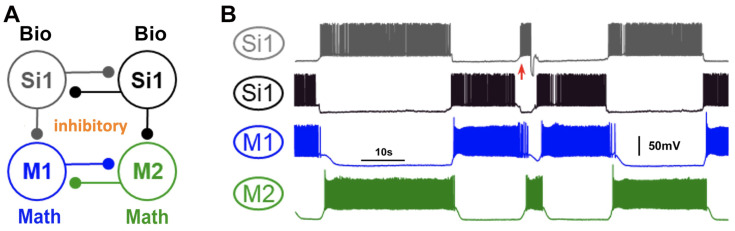
(**A**) A *blended* configuration utilizing the voltage recordings from biological *Melibe* interneurons (Si1L/R) is used to train, via unidirectionally inhibitory synapses, the tonically spiking mathematical neurons, which are also coupled reciprocally by inhibitory synapses in a curare bath, and under perturbation with a positive electric pulse injected in Si1L during its long quiescent phase, to switch the bursting order in the HCO. (**B**) A previously recorded voltage time series of the *Melibe* swim interneurons (Si1s), alongside the voltage time series of the two mathematical neurons. The inhibitory synapses are indicated by •, while the black synapses depict the mathematical synapses driven by the biological recordings from the Si1 neuron. The mathematical neurons are represented by M1 (in blue) and M2 (in green), while Si1 denotes the previously recorded swimming interneuron (in black). The perturbation location is highlighted with a red arrow.

**Figure 19 brainsci-14-00468-f019:**
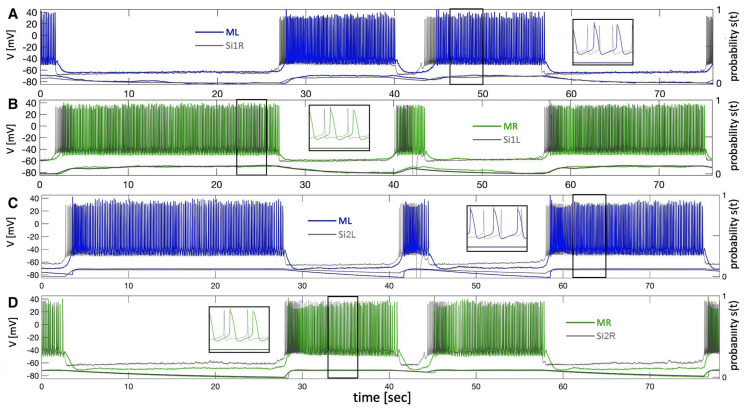
Time series of the voltage of mathematical neurons are overlaid on top of the neurophysiological voltage recordings of their images—biological neurons Si1 and Si2 in the blended system presented in Figure 17 and Figure 18, as well as simulated synaptic probabilities S(t) that agree rather well. (**A**,**B**) From Figure 18: Voltage time series (in blue and green) of the mathematical neurons, ML/R, superimposed on the voltage recordings of their corresponding neurons, Si1, right and left. resp., along with a zoomed-in section of an active burst indicated by a black box. (**C**,**D**) From Figure 17: the voltage time series (in blue and green) generated by the mathematical neurons are superimposed onto the voltage recordings of the corresponding Si2L/R cells, along with the simulated synaptic probabilities show a good agreement.

**Table 1 brainsci-14-00468-t001:** This table presents a vector array of voltage values sampled at a rate of 1.05 ms electrophysiological recordings on the *Melibe* swim CPG, in vitro. The time array is located in the left column, while the remaining columns represent voltage samples of interneurons 1L, 1R, and 3L. These recordings serve as the training data for blended and bio-math neural networks, which are coupled and run using such a dataset.

Time (s)	1L (mV)	1R (mV)	3L (mV)	3R (mV)
0	−52.14	−7.96	−36.16	−46.87
0.00105	−52.29	−20.14	−34.05	−46.97
0.00210	−52.15	−27.64	−33.26	−46.74
0.00315	−52.03	−32.23	−33.83	−46.51
0.00420	−53.87	−35.02	−33.92	−46.17
…	…	…	…	…

## Data Availability

The error function code is available in the GitHub repository at https://github.com/jbourahmahGSU/Voltage-recording-Error-function.git (accessed on 3 April 2024). An example of the blended system code that incorporates the *Melibe* CPG recordings with the mathematical Si models can be found at https://github.com/jbourahmahGSU/Blended-System.git (accessed on 3 April 2024). Samples of voltage recordings used in the paper are available upon request.

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
