# Peer review of "Error Function Optimization to Compare Neural Activity and Train Blended Rhythmic Networks"

_brainsci, 2024, doi:10.3390/brainsci14050468_

Round 1

Reviewer 1 Report

Comments and Suggestions for Authors

The authors have presented voltage recordings biological neurons to drive and train mathematical models, facilitating the derivation of the error function for further parameter optimization. The calibration process incorporates measurements such as action potential (AP) frequency, voltage moving average, voltage envelopes, and the probability of post-synaptic channels. The findings indicate that a weighted sum of simple functions is essential for comprehensively capturing a neuron’s rhythmic activity. The proposed methodology to study neural activity is interesting; however, it needs revisions from the point authors to improve the quality of the manuscript. For more details about the review report please find the attachment

Comments on the Quality of English Language

 Minor editing of English language required

Author Response

Please see the pdf attached 

Reviewer 2 Report

Comments and Suggestions for Authors

Remarks are in  the attached file

Author Response

Please see the pdf attached

Reviewer 3 Report

Comments and Suggestions for Authors

In this paper, the authors of the article introduced a new set of quantitative measures to facilitate the comparison between biological recordings and mathematical models in the case of electrophysiological experiments. The "blended" system approach offers an objective and efficient method for this comparison, utilizing voltage recordings of biological neurons to train mathematical models. Calibration involves various measurements, including action potential frequency and voltage envelopes. Using the sea slug Melibe leonina as a model circuit, experiments were conducted to evaluate the method's effectiveness. Results suggest that a weighted sum of simple functions is crucial for capturing neuron activity comprehensively. In conclusion, this approach shows promise for advancing research in neural circuitry by enabling objective and high-throughput comparisons between biological and mathematical models.

The paper is clearly written, with reference to the code used and an example, with adequate bibliography.

Author Response

We thank Reviewer 3 for his/her comments and suggestions.

Round 2

Reviewer 1 Report

Comments and Suggestions for Authors

Accept